# RETRACTED: Polymeric Solar Cell with 19.69% Efficiency Based on Poly(o-phenylene diamine)/TiO_2_ Composites

**DOI:** 10.3390/polym15051111

**Published:** 2023-02-23

**Authors:** M. Sh. Zoromba, M. H. Abdel-Aziz, A. R. Ghazy, N. Salah, A. F. Al-Hossainy

**Affiliations:** 1Chemical and Materials Engineering Department, King Abdulaziz University, Rabigh 21911, Saudi Arabia; mzoromba@kau.edu.sa; 2Chemical Engineering Department, Faculty of Engineering, Alexandria University, Alexandria 21544, Egypt; 3Physics Department, Faculty of Science, Tanta University, Tanta 31527, Egypt; ahmed.ghazy@science.tanta.edu.eg; 4Center of Nanotechnology, King Abdulaziz University, Jeddah 21589, Saudi Arabia; nsalah@kau.edu.sa; 5Chemistry Department, Faculty of Science, New Valley University, Al-Kharga 72511, Egypt; ahmed73chem@scinv.au.edu.eg

**Keywords:** mono nanocomposite, nanostructure thin film, TD-DFT method, optical properties

## Abstract

Conducting poly orthophenylene diamine polymer (PoPDA) was synthesized via the oxidative polymerization route. A poly(o-phenylene diamine) (PoPDA)/titanium dioxide nanoparticle mono nanocomposite [PoPDA/TiO_2_]^MNC^ was synthesized using the sol–gel method. The physical vapor deposition (PVD) technique was successfully used to deposit the mono nanocomposite thin film with good adhesion and film thickness ≅ 100 ± 3 nm. The structural and morphological properties of the [PoPDA/TiO_2_]^MNC^ thin films were studied by X-ray diffraction (XRD) and scanning electron microscope (SEM). The measured optical properties of the [PoPDA/TiO_2_]^MNC^ thin films such as reflectance (R) in the UV–Vis-NIR spectrum, absorbance (Abs), and transmittance (T) were employed to probe the optical characteristics at room temperatures. As well as the calculations of TD-DFT (time-dependent density functional theory), optimization through the TD-DFTD/Mol^3^ and Cambridge Serial Total Energy Bundle (TD-DFT/CASTEP) was employed to study the geometrical characteristics. The dispersion of the refractive index was examined by the single oscillator Wemple–DiDomenico (WD) model. Moreover, the single oscillator energy (
Eo
), and the dispersion energy (Ed) were estimated. The obtained results show that thin films based on [PoPDA/TiO_2_]^MNC^ can be utilized as a decent candidate material for solar cells and optoelectronic devices. The efficiency of the considered composites reached 19.69%.

## 1. Introduction

One of the most crucial approaches to meeting the rising global energy needs using a renewable resource is to directly harvest energy from sunlight using photovoltaic technology. Due to the possibility of fabricating polymeric solar cells (PSCs) onto large regions of lightweight flexible substrates by solution processing at a cheap cost, they represent a promising alternative for creating clean and renewable energy [1,2]. Organic photovoltaic cells with a single component active layer sandwiched between two electrodes with different work functions only had a very low power conversion efficiency because of insufficient charge carrier formation and unequal charge transport [3]. Due to their unique characteristics, polymer solar cells are regarded as a potential photovoltaic technology. For instance, all-polymer active layers in organic solar cells can have good stability and resilience together [4,5,6,7]. Due to a lack of effective polymeric materials, this form of photovoltaic has had a two-decade-long slow progress in terms of power conversion efficiency (PCE). The PCE of polymers has, however, reached a new level as a result of the recent use of polymerized small molecule acceptors. Multiple reports with PCEs of 10% [8,9,10,11,12,13] and >10% (over a short period of time) have been published [14,15,16,17]. The best performing polymers to date have PCEs in the 15–16% range [17,18,19,20,21,22,23,24]. However, there is still a long way to go for polymers compared to the successes of the small molecule-based solar cells (18% PCE) [25,26,27,28,29]. According to the knowledge gained from small molecule-based solar cells, optimizing the morphology and consequently the device characteristics is crucial for raising the efficiency of a material system that already exhibits respectable photovoltaic performance. It would be considerably more challenging in all-polymer solar cells since the morphology is formed by two materials that are intricately intertwined and both have long conjugated chains. In the case of polymer solar cells with advanced efficiency, it would be particularly difficult to improve all three major device parameters simultaneously, namely open-circuit voltage (VOC), short-circuit current density (JSC), and fill factor (FF), and one improvement is typically accompanied by the decline in the other (s). However, this does not imply that from the standpoint of device engineering, this cannot be carried out. For instance, as many have noted, the introduction of additives (solvents or solids, tiny compounds, or polymers) has shown to be a successful tactic for enhancing the photovoltaic performances of solar cells [30,31,32,33,34,35]. In accordance with the photovoltaic working mechanism, most organic photodetectors described to date have external quantum efficiency (EQE) values below one. Consequently, the low EQE of photodiode-type photodetectors restricts their possible applications. The group led by Yokoyama was the first to disclose a substantial photomultiplication (PM) phenomena based on organic pigment layers sandwiched between two electrodes. This phenomenon was rationally explained in terms of electron tunnelling injection produced by the concentration of photogenerated holes along the interface of the organic layer and the metal electrode [36].

In the last 10 years, many reports have exhibited the crucial role of the contemporary TD-DFT applications (DMol^3^ and CASTEP methodologies) in investigating the structure and phase stability [37,38,39]. The comprehensive energy-based technique has come a long way in a very short time; however, there are massive obstacles to overcome for its application in estimating and investigating spectroscopic characteristics. The following report demonstrates the calculation of the linear and nonlinear optical spectra, XRD, vibrational infrared spectra, and spectroscopic difficulties utilizing the restricted programming language [40,41,42]. The goal is to demonstrate how similar atomistic modeling techniques can be used consistently to attain high levels of accuracy throughout the experimental research [43,44]. Applying an initial pseudopotential allows one to determine the potential of an electron-ion in formulations that are either ultra-soft or standard-conserving. The conscience-consistent approach, corresponding charge intensity, and Kohn–Sham wave functions are all determined using the direct energy minimization results that were obtained. There are explicit methods for combining gradients with density mixing. The form that can describe systems with a finite number of variables is determined by the strong DFT electron. [45,46,47]. This work was designed not only for enhancing the performance of synthesized and fully structurally characterized [PoPDA/TiO_2_]^MNC^ nanostructured thin films for inserting into the devices such as solar cells. A significant role is also the control of the defects and impurity states in the organic molecular crystals to achieve proper photoelectrical characteristics. Moreover, [PoPDA/TiO_2_]^MNC^ is structurally characterized by XRD. The thin film of [PoPDA/TiO_2_]^MNC^ is fabricated by deposition on top of glass using (PVD) with annealing at 298 K for 1 h, and then, its optical characteristics investigated. In addition, the experimental data are investigated by dispersion models for the real refractive index and extinction coefficient. Moreover, the role of doping on optical properties is demonstrated.

## 2. Experimental Part

### 2.1. Raw Materials

None of the compounds were further purified before usage. Ortho phenylenediamine was obtained from Across Organics, Thermo Fisher Scientific. The following substances were used as purchased: ferric chloride (Aldrich), ethanol, anhydrous dimethyl formaldehyde (DMF), dimethyl sulfoxide (DMSO), and polyethylene glycol (PEG200) from Shanghai Chemicals (Shanghai, China). Sigma-Aldrich provided us with hydrochloric, hydrofluoric, nitric, and acetic acids as well as a single crystal of p-Si and tetrahydrofuran (THF) and titanium dioxide (TiO_2_).

### 2.2. Synthesis of PoPDA Polymer

Typically, 80 mL of ethyl alcohol was used to dissolve 8.64 g of o-phenylenediamine (oPDA). Then, 4 mL of PEG200 was added to the oPDA solution (as a soft surfactant to control the shape of the resultant polymer). After that, 6–7 mL of concentrated HCl was added to the oPDA solution while it was being stirred with a magnetic stirrer at 650 rpm and 25 °C. Under the previously mentioned circumstances, 160 mL (0.5 M) anhydrous ferric chloride solution was then gradually added to the oPDA solution as an oxidizing agent. It was discovered that the resulting solution had a pH of 0. The monomer and initiator are at a 1:1 ratio. The PoPDA that was produced was for seven days at room temperature. To remove of the extra oxidizing agent, surfactant, and protonated monomer, the resultant PoPDA was repeatedly rinsed with distilled water. At 60 °C, the resultant PoPDA was dried.

### 2.3. Manufacturing Techniques of Au/[PoPDA/TiO_2_]^MNC^/p-Si/Al Heterojunction Diodes

In total, 0.5 g of the powder (PoPDA) was dissolved in 80 mL tetrahydrofuran (THF) under stirring at 65 °C for 35 min in tightly closed beakers under magnetic stirrer for additional 35 min at room temperature. To prepare nanocomposite films, the [PoPDA/TiO_2_]^MNC^ solution was doped with x = 3.0, 5.0, and 10.0 wt% of the sol–gel-prepared [TiO_2_]^NPs^. The required mass (
wtTiO2
) of [TiO_2_]^NPs^ fillers was determined utilizing the following equation:
(1)
x wt%=(wtTiO2wtTiO2+0.5)×100

where 0.5 in the denominator is the total mass of the homopolymer [PoPDA]. The 
wtTiO2
 was dissolved in 10 mL THF using ultrasonic homogenizer and then added to the blend solution under stirring process. The homopolymer [PoPDA] solution was put in Petri dishes and left for one day to evaporate the solvent completely. The manufacturing of [PoPDA/TiO_2_]^MNC^ thin films involved the use of physical vapor deposition (PVD). Thin films were deposited onto a single crystal of (p-Si) wafer and an ITO/glass substrate using a UNIVEX 250 Leybold (Germany) at a base pressure of 5 × 10^3^ Pa, interdigitated electrodes spaced by 75 m, and a deposition rate of 3 
Å
/s. The thin films of nanostructure [[PoPDA/TiO_2_]^MNC^ were balanced using a quartz crystal microbalance on a UNIVEX 250 Leybold machine (as shown in Figure 1a,b). In contrast to other investigations [48,49], the vacuum pressure employed in this investigation was 5 × 10^3^ Pa, compared to 27 × 10^2^ and 27 × 10^1^ Pa. There were no compositional changes that suggested substrate reactivity from the substrate contact to the film surface. Additionally, based on the results, [PoPDA/TiO_2_]^MNC^ nanocomposite thin films were not made with a change in vacuum pressure.

### 2.4. Application of TD-DFT/Mol^3^ and TD-DFT/CASTEP Technique

In order to examine the frequency studies and molecular structure performance for the polymer [PoPDA]^Iso^ and nanocomposite [PoPDA/TiO_2_]^Iso^ in the TD-DFT in gas phases state estimated evidence of PBE/GGA functionality, natural pseudo-positive preservatives, and a straightforward DNP set for acceptable compounds, DMol^3^ was used [50,51]. The plane-wave power cut-out was calculated to have a total value of 830 eV. For instance, the spectroscopic and physical properties of the polymer [PoPDA]^Iso^ and nanocomposite [PoPDA/TiO_2_]^Iso^ in gaseous state were detected using DMol^3^ IR features, leading to a GP frequency approximation. Three more factors have been found to alter how Becke functions [52]. B3LYP Lee Yang Parr [53] Form and vibrant regularity (IR), the polymer [PoPDA]Iso, the nanocomposite [PoPDA/TiO_2_]^Iso^, and the nanofluid in the gaseous phase have all been improved with WBX97XD/6-311G. The symmetric variables, images of enhancements, power, and vibration of processed nanocomposite mixtures are examined by the GAUSSIAN 09W system programming. Numerous useful details about the relationship between the setup and the variety of results offered by our group have been revealed by the WBX97XD/6-311G B3LYP technique [54]. The portrayals of polymer [PoPDA]^Iso^ and nanocomposite [PoPDA/TiO_2_]^Iso^ Gaussian in isolated molecules, deliberate variations in descriptors, prototype vitality data, and the employment of various modifications with varying difficulties were evaluated using the GAUSSIAN 09W and DMOl^3^ methodologies.

### 2.5. Characterization

The various methods of analysis and typical settings that describe the polymer [PoPDA]^TF^ and nanocomposite [PoPDA/TiO_2_]^MNC^ were recorded by utilizing the following instruments: the powder X-ray diffraction (XRD) analysis at 2θ = 4–90°, its collected data for the line broadening on the synthesized powder and phase composition by using a Shimadzu apparatus with Xpert diffractometer (Rigaku D-max C III, X-ray diffractometer using Ni-filtered Cu Kα radiation at λ = 1.5418 Å). The surface morphology of the produced [PoPDA]^TF^ and [PoPDA/TiO_2_]^MNC^ thin films were studied utilizing a scanning electron microscope (SEM) (QUANTA FEG 250) and EDX unite. The TEM images were captured at an accelerating voltage of 120 kV utilizing a JEM 200CX transmission electron microscope (JEOL, Tokyo, Japan). The optical properties and absorbance of the organic compounds were computed on a PerkinElmer Lambda 365 spectrophotometer. Photoluminescence was measured with a Hitachi F-7100 fluorescence spectrometer equipped with a detector photomultiplier R928F. Current density versus voltage (J–V) curves of the [PoPDA/TiO_2_]^MNC^ were measured by a Keithley 2400 source meter in air conditions that scanned from 300 nm to 700 nm, under the 100 ± 3 nm illuminations with intensities of 300 mW/cm^2^, 100 mW/cm^2^, and 50 mW/cm^2^, respectively. The monochromatic light used in all these measurements was provided by a 150 W xenon lamp coupled with a monochromator. EQE is calculated as:
(2)
R=Jph/Iin

and

(3)
EQE=Rhν/e

where *R* is the responsivity, 
Jph
 is the photocurrent density, 
Iin
 is the intensity of incident light, e is absolute value of electron charge, and hu is the energy of incident photon.

## 3. Results and Discussion

### 3.1. X-ray Powder Diffraction (XRD) for [PoPDA]^Iso^ and [PoPDA/TiO_2_]^Iso^ Thin Film

X-ray diffraction pattern (XRD) studies were applied to determine the crystalline phase configurations and atomic orientation of thin film and multi-layered films. Experimental XRD studies for [PoPDA]^TF^ and [PoPDA/TiO_2_]^MNC^ thin film are shown in Figure 2a,b. The XRD patterns for [PoPDA]^TF^ and [PoPDA/TiO_2_]^MNC^ thin films have strong diffraction peaks in the range 10° ≤ 2θ ≤ 60° and 20° ≤ 2θ ≤ 60°, respectively, which indicate the long-range and crystallinity order. The estimated crystalline structure of [PoPDA]^TF^ and [PoPDA/TiO_2_]^MNC^ thin films, crystal device variance, hkl, *d*-spacing (d), and FWHM (β) are shown in Table 1. The XRD patterns in Figure 2a show a polycrystalline structure with an ORTHORHOMBIC space group unit cell for [PoPDA]^Iso^ thin film with the parameters a = 13.95(6) Ǻ, b = 13.83(3) Ǻ, c = 9.943(9) Ǻ, α = 90°, β = 90°, γ = 90°, and volume = 1910(8) Ǻ^3^ [55]. Figure 2b shows the XRD patterns of [PoPDA/TiO_2_]^Iso^ thin film with a MONOCLINIC space group unit cell with the parameters a = 13.342(1) Ǻ, b = 13.127(2) Ǻ, c = 8.994(2) Ǻ, α = 90°, β = 100.28°, γ = 90°, and volume = 1550.16(2) Ǻ^3^. The [PoPDA]^TF^ and [PoPDA/TiO_2_]^MNC^ thin films’ average crystalline sizes (D_Av_) were found to be 102.5 and 88.7 nm, respectively (Table 1). It is known that the average crystallite size and size distribution strongly affect the characteristics of the semiconducting nanomaterials, especially for broad distribution spectrum nanoscale crystallites. The full width at half maximum FWHM (*β*) and crystalline size D were measured in the range 10°≤ 2θ ≤ 60° for [PoPDA]^Iso^ and listed in Table 2 using Debye–Scherrer:
(4)
DAv=0.9λβcosθ

where *λ* = 0.154 nm and *β* is in radians. The database code amcsd 0001206 is in excellent agreement with the interplanar distance d-spacing. Peak lines estimated by the TD-DFT-DFT and Crystal Sleuth Microsoft applications are close to the measured data [56].

### 3.2. Geometric Study of [PoPDA]^Iso^ and [PoPDA/TiO_2_]^Iso^ Isolated Molecules

The chemical and physical features of the gaseous phase of the polymer [PoPDA] and nanocomposite [PoPDA/TiO_2_] as isolated molecules were investigated using electron density and electrostatic potential as shown in Figure 3a–c [59,60].

In TD-DFT and TD-DFT/Gaussian ideas, the electron density was used as a crucial operator to explore the isolated electron system for [PoPDA]^Iso^ and [PoPDA/TiO_2_]^Iso^ (Figure 3a). Figure 3b shows the outstanding potential expansion of the gaseous phases of [PoPDA]^Iso^ and [PoPDA/TiO_2_]Iso. The molecular electrostatic potential (MEP), which was calculated based on surface density, increased the likelihood of electron transfer [PoPDA]^Iso^ and [PoPDA/TiO_2_]^Iso^ in the gaseous phase.

Figure 3c shows 3D pictures of the MEP for single molecules of electron transfer, with the favorable regions for electrophilic and nucleophilic assaults designated in red and blue, respectively. Accordingly, for [PoPDA]^Iso^ and [PoPDA/TiO_2_]^Iso^, based on the data in Figure 3c, the MEP ranges were −1.820 × 10^−1^ ≤ [MEP] ≤ 8.608 × 10^−2^ and −2.072 × 10^−1^ ≤ [MEP]≤ 2.491 × 10^−1^ for [PoPDA]^Iso^ and [PoPDA/TiO_2_]^Iso^, respectively, in the isolated phase of the molecule. The MEP diagram also revealed the potential negative regions of the hydrogen atoms’ positive potential. Red, brown, and blue were listed as the order of the colors [60,61]. Blue introduced the strongest electron attraction, while red suggested the largest electron repulsion [62,63] Additionally, the long pair of electronegative atoms was aligned with the negative regions. The molecular MEP investigation demonstrated that nitrogen and the -bond in aromatic ring molecules were introduced as negative areas. The attack sites of the probable nucleophiles ic can be assumed to be the extremely positive locations where deprotonated ethylene (CH_2_=CH–) was represented in [PoPDA]^Iso^ and [PoPDA/TiO_2_]^Iso^, with a maximum value of +3.87 a.u. According to the MEP map computation, the sites with positive potential were (–CH_2_C–), while the sites with negative potential were (–C=N), (CH_2_=CH–), and (–CH_2_C–). Molecule intermolecular interaction has given remarkably evidence for these locations. Consequently, intermolecular hydrogen-bonding existence is emphasized in Figure 3c.

Considering the advantage of the *TD-DFT/DMOl^3^* procedure, 
ΔEgOpt
 values were measured depending on the contradiction between the *LUMO* (lowest unoccupied molecular orbital) and *HOMO* (highest occupied molecular orbital) states as presented in Figure 3a. Basically, the *LUMO* and *HOMO* states impact the fragment molecular orbital (FMO) method’s complicated analysis in quantum chemical simulations. The computed energy values of 
EHOMO
, 
ELUMO
, and 
ΔEgOpt
 are presented in Table 2. In addition, equations such as

(5)
μ=(EHOMO+ELUMO)/2


(6)
η=(ELUMO−EHOMO)/2


(7)
S=1/2η


(8)
ω=μ2/2η


(9)
σ=1/η


(10)
ΔNmax=−μ/η

were employed to determine the values of the softness (
σ
), chemical potential (*μ*), global hardness (
η)
, global softness (*S*), the global index of electrophilicity (ω), the electronic charge maximum amount (
ΔNmax
), and electronegativity (χ) [64,65] of [PoPDA]^Iso^ and [PoPDA/TiO_2_]^Iso^

The negative 
EHOMO
 and 
ELUMO
 energies introduced stability into isolated molecules. Additionally, by employing the crucial quantum chemical property (ω), the stability of energy with extra electronic charge received through the device was evaluated [66].

The results of the M062X/6-31 + G(d,p) computations were found for the HOMO and LUMO in a gaseous ground state. Additionally, utilizing the energy difference between fragment molecular orbitals theory, the molecule state of equilibrium, which is important for determining the grasping electricity transit, and electrical conductivity were estimated (FMOs). Entropy levels that are entirely negative demonstrate the stability of isolated molecules [67]. The obtained results of the FMO method indicated electrophilic sites of aromatic compounds. Moreover, when dimer molecule bonds (DMB) grew and bond length diminished, HOMO energy (
EH
) was boosted utilizing the variance technique of the Gutmannat on DMB sites [68]. Furthermore, considering the energy gap (
EgOpt
) optimization, the molecular system stability and reactivity characteristics were estimated. Meanwhile, the stability and responsiveness were evaluated according to the hardness and softness [69,70]. In the following equation

(11)
x=(EH+EL)/2

the electronegativity was calculated and the energy bandgap that exhibits the connection between the charge transport in the molecule is introduced in Table 2.

### 3.3. SEM Morphology

SEM images of the [PoPDA]^TF^ and [PoPDA/TiO_2_]^MNC^ thin films are shown in Figure 4a,b, respectively, at high and low magnifications. Uniform microrods with a diameter of 12 μm and different rod lengths are shown for [PoPDA]^TF^ in Figure 4a. A high magnification image of the [PoPDA/TiO_2_]^MNC^ surface is shown in Figure 4b demonstrating the good morphological features of physical vapor deposition, which are similar to pulsed-laser-deposited film [71]. The examined film surface shows the development of a dense surface, which is related to the impact of inorganic precursors on the structure of the [PoPDA/TiO_2_]^MNC^ film, as well as the rates of relative evaporation and condensation, in which the volume and size of the pore were determined [72].

Generally, the variety of nanoparticle shapes that can be used is an important detail in the composition of hybrid solar cells. A specific shape of nanoparticles may improve the efficiency and performance of hybrid solar cells, but no direct relationship has been demonstrated. Lazaro A. et al. [73], for example, investigated the role of the PbSe nanostructure shape by comparing the elongated or flaky shape and the spherical nanoparticle shape. They discovered that the high aspect ratio of elongated particles improves light absorption efficiency when compared to spherical nanoparticles [73]. The current work demonstrates that the power conversion efficiency of a solar cell is dependent on the conjugation of PoPDA for light harvesting and the shape particles of PoPDA (rods as seen in the SEM image), with electron transfer being easier through rods than through spherical shape particles [8,74].

### 3.4. AFM Analysis

Atomic force microscope images were used to study the structural morphologies and surface topography of [PoPDA]^TF^ and [PoPDA/TiO_2_]^MNC^ (Figure 5 and Figure 6). The [PoPDA]^TF^ AFM images in Figure 5 show rectangular microrods with different rod lengths and average face lengths of 50 nm. The 3D AFM images in Figure 5 were used to study the surface topography of [PoPDA]^Iso^; the roughness studies show the root mean square roughness ranges between 1.935 and 2.422 with a maximum height of 11.134 and 12.625 Ǻ/nm and an average height of 5.58 and 6.306 for two different stud areas of the [PoPDA]^TF^ thin film. The [PoPDA/TiO_2_]^MNC^ thin film AFM images show irregular surface block rods, while for the roughness studies on the AFM 3D images of [PoPDA/TiO_2_]^MNC^, Figure 6 shows root mean square roughness ranges between 1.918 and 2.487 with a maximum height of 11.04 and 13.738 Ǻ/nm and an average height of 5.523 and 6.916 for two different study areas of the [PoPDA/TiO_2_]^MNC^ thin film. The similarity in the roughness studies between [PoPDA]^TF^ and [PoPDA/TiO_2_]^MNC^ indicates that the roughness of the surface is because of the nature of PoPDA and TiO_2_ has a negligible effect on it.

### 3.5. Optical Properties

UV–Vis absorption spectroscopy gives important information about the electron transition between the highest occupied and lowest unoccupied molecular orbits (HOMO and LUMO). As shown in Figure 7a, the absorption spectra of [PoPDA]^TF^ show two absorption peaks at 332 nm and 387 nm, which are assigned to the π-π* transition and quinoid imine unit’s existence (–C=N–) [75,76], while the absorption spectra of [PoPDA/TiO_2_]^MNC^ show a disappearance of the absorption peak at 387 nm and a new absorption peak appears at 480 nm, which belongs to TiO_2_ nanoparticles. The addition of TiO_2_ nanoparticles results in a reduction in the 332 nm absorption peak intensity to 33.33% which agrees with the previous work [77]. The CASTEP optical properties in Figure 7b show strong agreement between the experimental and simulated data with minor differences.

Using absorption coefficient α, which was calculated using the relation: 
α=2.303A/x
, where x is the film thickness, the minimum energy required to move the electron from valence to conduction band, which is known as the optical energy gap (
Egopt
), can be calculated using Tauc’s relation:
(12)
(αhυ)1γ=B(hυ−Eg)

where 
hυ
 is the photon energy and γ = ½ for direct transition and 2 for indirect transition. Figure 8 shows the relation between 
(αhυ)1γ
 and photon energy 
hυ
 for direct and direct transitions. The values of the optical energy gap 
Egopt
 for [PoPDA]^TF^ and [PoPDA/TiO_2_]^MNC^ are 2.296 eV and 2.114 eV, respectively, in the case of direct transition, and 3.511 eV and 3.209 eV, respectively, in the case of indirect transition. The calculated values of the direct and indirect energy gaps show a significant decrease in the direct and indirect energy gap values by adding TiO_2_ nanoparticles and indicate the preference for direct electron transition.

Interaction between incident light and polymer nanocomposites plays an important role in using polymer nanocomposites in different applications such as optical devices and solar cells. The refractive index n(λ) and absorption index k(λ) indicate how much incident light is refracted and absorbed when interacting with the thin film. The absorption index k(λ) and refractive index n(λ) were calculated using the equations:
(13)
k=αλ4π 


(14)
n=(1+R1−R)+4R(1−R)2−k2


Figure 9a shows the variance of n(λ) and k(λ) as a function of photon energy; the refractive index n(λ) values ranged between 1.72 and 1.31 for [PoPDA]^TF^ and decreased to ranges between 1.63 and 1.29 for [PoPDA/TiO_2_]^MNC^, which indicates the improvements in the optical properties.

On the other hand, the similarity in the behavior between the absorption index k(λ) and absorption spectra in Figure 9a indicates the homogeneity of the film since there is no scattered light observed. The simulated refractive index n(λ) and absorption index k(λ) found using the CASTEP optical properties software are shown in Figure 9b and show a similar behavior to the experimental data.

Real and imaginary parts of the dielectric constant ε_1_ and ε_2_ represent the material’s ability to store electric energy and the ohmic resistance of the material. Depending on the refractive index n(λ) and absorption index k(λ) values, the real and imaginary parts of the dielectric can be calculated as

(15)
ε1=n2−k2

and

(16)
ε2=2nk


Figure 10a shows ε_1_(ω) and ε_2_(ω) as a function of photon energy, and similar behavior of the simulated ε_1_(ω) and ε_2_(ω) is observed (Figure 10b). For [PoPDA]^TF^, ε_1_(ω) reaches its maximum intensity of 2.961 (F·m^−1^) at hν = 3.725 eV, while ε_2_(ω)’s maximum intensity was 4.49 × 10^−8^ (F·m^−1^) at the same photon energy. On the other hand, ε_1_(ω) and ε_2_(ω) reached their maximum intensities of 2.667 and 4.309 × 10^−8^ (F·m^−1^), respectively, at hν = 2.569 eV for [PoPDA/TiO_2_]^MNC^. High values of ε_1_(ω) for both [PoPDA]^TF^ and [PoPDA/TiO_2_]^MNC^ indicate a great capacity to store electric energy, while low values of ε_2_(ω) indicate a good electrical conductivity behavior. DFT-CASTEP simulations of dielectric constants show a good agreement with dielectric experimental data.

The electromagnetic wave response by the substance is explained by the optical conductivity σ(ω). Real optical conductivity σ_1_(ω) and imaginary optical conductivity σ_2_(ω) can be calculated in terms of ε_1_(ω) and ε_2_(ω) as:
(17)
σ1(ω)=εoε2ω


(18)
σ2(ω)=εoε1ω

where 
εo
 is the free space permittivity and 
ω
 is the angular frequency.

Figure 11a shows the variations in σ_1_(ω) and σ_2_(ω) as a function of photon energy, and similar behavior of σ_1_(ω) and σ_2_(ω) is shown for [PoPDA]^TF^, while for [PoPDA/TiO_2_]^MNC^, σ_2_(ω) has a different behavior than σ_1_(ω), which is a result of adding TiO_2_ nanoparticles. Simulated data using DFT-CASTEP simulations in Figure 11b are found to strongly match the experimental data.

The Wemple–DiDomenico model is applied on [PoPDA]^TF^ and [PoPDA/TiO_2_]^MNC^ in Figure 12 in order to determine dispersion energy E_d_ and single oscillator energy E_o_ (related to direct energy gap by the relation E_o_ ≈ 1.5 → 2 E_g_) using the following equation [78]:
(19)
(n2−1)−1=EoEd−1EoEd(hυ)2


Dispersion energy E_d_ and single oscillator energy E_o_ are calculated and listed in Table 3. Using the equation

(20)
n∞2−1/n2−1=1−(λo/λ)2

the long-distance refractive index (
n∞
) and average inter-band oscillator wavelength (
λo
) were calculated for [PoPDA]^TF^ and [PoPDA/TiO_2_]^MNC^ (Figure 12b), while the average oscillator strength was calculated from the relation

(21)
So=n∞2−1/λo2


By applying the Sellmeier single term oscillator model [79,80]

(22)
n2=εl−(e2N/4π2c2m*)λ2

the lattice dielectric constant (
εl
) and the carrier amount ratio to the electron effective mass (
e2/πc2
)
(N/m*)
 were calculated (Figure 12c). 
n∞
, 
λo
, 
So
, 
εl,
 and (
e2/πc2
)
(N/m*)
 are calculated and listed in Table 3.

The comparative studies for the nanoblend [PoPDA]^TF^ and nanocomposite [PoPDA/TiO_2_]^MNC^ are represented in Table 4. The refractive index n(λ) and direct and indirect energy gaps 
(ΔE)
 values for all the nanoblend and nanocomposite are given in Table 4.

### 3.6. Laser Photoluminescence Behavior

Photoluminescence spectra for [PoPDA]^TF^ and [PoPDA/TiO_2_]^MNC^ thin films in the range of 1.5 to 6.25 eV are shown in Figure 13. The first investigations show that for [PoPDA]^TF^, the main emission peak appeared at 2.469 eV; this peak is related to the excess of fluorophore molecules in the polymer chain. In the case of [PoPDA/TiO_2_]^MNC^, the emission peak, in blue, shifted to 3.297 eV; this shift is due to the addition of TiO_2_ nanoparticles. A shift of 0.828 eV occurred when TiO_2_ nanoparticles were added to the PoPDA polymer, which is a fluorescent polymer due to the random lasing in the weakly scattering system. In random lasing cases, the feedback does not occur in conventional resonant cavities, but the introduction of nano-scatterers (TiO_2_ nanoparticles in this case) as a disordering factor strongly influences the feedback mechanism [82,83].

Commission Internationale de l’Eclairage (CIE) graphs (Figure 14) illustrate the emission colors of [PoPDA]^TF^ and [PoPDA/TiO_2_]^MNC^, for [PoPDA]^TF^ digital photographs obtained at hν = 2.469 eV, which is located at the coordinates (0.32, 0.27), while the digital photographs for [PoPDA/TiO_2_]^MNC^ were obtained at hν = 3.297 eV and located at the coordinates (0.33, 0.34). A significant change in the color of the emission can be figured out since the emission color changed from light purple for [PoPDA]^TF^ to pale blue for [PoPDA/TiO_2_]^MNC^. The effect of adding TiO_2_ to PoPDA to fabricate [PoPDA/TiO_2_]^MNC^ results in making it close to emitting white light from a single material, giving it potential applications in lighting and display devices.

### 3.7. Electrical Properties

#### 3.7.1. The Influence of Applied Potential Difference (V) on the Current (I)

Dark and illuminated J-V characteristic curves for the Au/[ PoPDA/TiO_2_]^MNC^/n-Si/Al heterojunction diode are shown in Figure 15 at 298° and different illuminations (50, 100, and 200 mW/cm^2^). As shown in Figure 15, V_oc_ changed from 0 V in the dark condition to 0.135 V in the illuminated condition; at a bias of +2 V, the device at different illumination intensities of 0, 50, 100, and 200 mW/cm^2^ showed forward currents of 0.001, 0.166, 0.667, and 1.828 mA/cm^2^, respectively. It is known that the quality of the diode is determined by the ratio of forwarding current to reverse current (J_F_/J_R_); the forward currents of the Au/[ PoPDA/TiO_2_]^MNC^/n-Si/Al heterojunction diode at +2 V are 0.001, 0.166, 0.667, and 1.828 mA/cm^2^, respectively, while the reverse currents at V = −2 V are 0.0044, 0.46, 1.4, and 5.8 μA/cm^2^, respectively, for the illuminations 0, 50, 100, and 200 mW/cm^2^, respectively. The resulting rectification ratios (J_F_/J_R_) are 0.22, 0.36, 0.47, and 0.31 × 10^3^, respectively, at ±2 V.

The nonlinear coefficient parameter r can be deduced using the relation: 
I=RVr
, where r is a constant, and the slope of these curves at different temperatures gives the value of r. I-V curves at different temperatures can be divided into two areas, one when V has small values and the slope is denoted as r_1_, the other one at large values of V and the slope is denoted as r_2_; we should take into account that r_2_ > r_1_. I–V characteristics of nonohmic behavior are confirmed by the values of r_1_ and r_2_, which are known as the nonlinear coefficient parameters recorded in Table 5. It can be figured out that the values of r_1_ are less than 2 and decrease with increasing temperature T (K) for Au/[ PoPDA]^TF^/n-Si/Al and Au/[ PoPDA/TiO_2_]^MNC^/n-Si/Al heterojunction diodes. Moreover, the values of r_2_ take the same behavior as r_1_ with temperature, the values of r_2_ decrease by raising T (K), and the polymers’ conduction mechanism is stated by the nonlinear coefficient parameters (r). When r = 1 we can obtain the ohmic behavior, and r = 2 (a dominant mechanism) indicates the incomplete trap-free space charge. Finally, the case when r > 2 indicates the defectiveness of the mechanism in terms of the trap charge.

#### 3.7.2. The Effect of Concentration and Heating on DC Conductivity

The nanoparticles’ uniform dispersion in polymer nanocomposites results in improving the polymer nanocomposites’ electrical characteristics. Since TiO_2_ nanoparticles interact with the polymer by a weak interfacial contact and TiO_2_ nanoparticles aggregate as a result of van der Waals force, uniform dispersion is then difficult to achieve. In this study, TiO_2_ nanoparticles dispersed excellently in the polymer matrix as indicated by the SEM and AFM images. Figure 16 shows the direct conductivity as a function of temperature T (K) at a constant voltage of 20 V [84]. The charge transfer was enhanced once the TiO_2_ nanoparticles interacted with PoPDA in the nanocomposite film. The σ_dc_ of the [PoPDA/TiO_2_]^MNC^ thin film was 66.06 × 10^−5^ S m^−1^, which is much larger than the value of 10.09 × 10^−5^ S m^−1^ for the [PoPDA]^TF^ thin film. The values of Ln σ_dc_ ranged between (−7.49 and−8.79) at 1/T = 0.0026 1/K and (−3.18 and −3.95) at 1/T = 0.002359 1/K. There were two problems that occurred in order to increase TiO_2_ nanoparticles’ loading: (1) the viscosity of PoPDA, which makes it difficult to absorb additional TiO_2_ nanoparticles and (2) the high surface energy of TiO_2_ nanoparticles, which makes them tend to agglomerate. An appropriate applied electrical field (E) tends to align TiO_2_ nanoparticles, which results in an agglomeration decrease and the network expands from the negative to the positive electrode. Conducting pathways are generated by following the stages: (1) the rotating of TiO_2_ nanoparticles to a specific angle according to the applied E, resulting in dipole moment generation at TiO_2_ nanoparticles edges, which makes them align to the electric field direction, (2) a 3D network is created by the attraction between the TiO_2_ nanoparticles until they make contact, and (3) the moving and adherence of TiO_2_ nanoparticles to the negative electrode. In conclusion, by ignoring the ionic conductivity, σ_dc_ of the nanocomposite films is primarily caused by the electronic conductivity of the TiO_2_ nanoparticles. The charge transfer is increased by increasing the temperature T (K), as seen in Figure 16. In the case of [PoPDA]^TF^ and the low concentrations of TiO_2_ nanoparticles, there were no 3D networks, but they would be activated by the charge carrier collected energy to leap potential barriers. Finally, heating and increasing the TiO_2_ nanoparticles’ content will optimize these routes, increase σ_dc_, and develop networks [49].

#### 3.7.3. Photovoltaic Properties of Au/[ PoPDA/TiO_2_]^MNC^/p-Si/Al Heterojunction Diode Films

Excited state photo-physics generated from absorption in the aggregate state can be investigated easily from the photovoltaic response. Moreover, increasing the nanoparticle content (TiO_2_) in the polymer nanocomposite results in an increase in the intensity and red shifting of the absorption band. Different light intensities, 50 mW/cm^2^ < P_in_ < 200 mW/cm^2^, were used to perform photovoltaic characteristics of the Au/[PoPDA/TiO_2_]^MNC^/p-Si/Al polymer solar cell. I-V curves of the solar cell were measured by a computerized Keithley 2635A system. A GPIB/USB cable was used to connect the source meter to the host computer. The photovoltaic performance, saturation current density (J_SC_), and open-circuit voltage (V_OC_) of the manufactured solar cell are listed in Table 6. It can be figured out that (J_SC_) and (V_OC_) were increased by increasing the incident light intensity. A prominent photovoltaic response was exhibited from the Au/[ PoPDA/TiO_2_]^MNC^/p-Si/Al heterojunction device since many high rectification degrees were exhibited in J-V characteristics (Figure 15). The saturation current density J_sc_ is related to the incident light intensity by the relation:
(23)
Jsc=APinγ

where A is a constant. The maximum current density (*J_m_*), voltage density (*V_m_*), and maximum power (*P_m_*) are tabulated in Table 5 for the different light intensities (P_in_). It can be figured out that *J_m_*, *V_m_*_,_ and *P_m_* are increased by increasing the light intensity. The fill factor (*FF*) and power conversion efficiency of the Au/[ PoPDA/TiO_2_]^MNC^/p-Si/Al heterojunction device were calculated using the equations:
(24)
FF=VmJm/VOCJSC

and

(25)
η=VOCJSC/Pin×FF×100

where *FF* and 
η
 values are listed in Table 5. As shown, the efficiency of the Au/[ PoPDA/TiO_2_]^MNC^/p-Si/Al heterojunction device is improved by increasing the light intensity. At P_in_ = 200 mW/cm^2^, FF and 
η
 were determined to be 68.04% and 19.69%, respectively [85,86].

For the devices, nanoblend Au/[PoPDA]^TF^/p-Si/Al (A) and nanocomposite Au/[PoPDA/TiO_2_]^MNC^/p-Si/Al (B), more electrons can be easily injected from P-Si onto the LUMO of the nanoblend and nanocomposite due to the small injection barrier of 0.4 eV and the large contact interface between P-Si and the nanoblend and nanocomposite. The injected electrons can be efficiently transported in the active layer along the continuous electron transport channels due to the nanoblend and nanocomposite, resulting in the relatively high dark current and high withstanding reverse bias up to 19 V. In order to investigate the photo response of all the polymer photodetectors (PPDs) with the nanoblend and composite, EQE spectra of all PPDs were measured under the given reverse biases and are shown in Figure 17. It is apparent that the EQE spectral shape of device A for the nanoblend is distinctly different from that of device B, as shown in Figure 17. The EQE values of device (A) are much larger than 100% in the spectral range from about 350 nm to 650 nm. However, the EQE values of device (B) are lower than 100% in the whole spectral range. Therefore, the PPDs for the nanoblend and nanocomposite can be classified as two different types: photodiode-type PPDs with EQE values lower than 100% and PM-type PPDs with EQE values higher than 100%. According to the energy levels of the nanoblend or nanocomposite, the isolated polymer aggregations in the blend films can be considered as electron traps due to the energy barrier of 1.3 eV between the LUMOs of the nanoblend and nanocomposite. The electrons trapped in the polymer near the Al cathode can build up a Coulomb field (i.e., energy level curved) to assist the hole tunneling injection from the Al cathode onto the HOMO of [TiO_2_]^NPs^ under reverse bias. An interesting phenomenon is that there is a distinct dip from about 490 nm to 570 nm and two apparent peaks at about 380 nm and 625 nm in the EQE spectra of a trap-assisted photomultiplication (PM)-phenomenon-type PPD. It is very apparent that the EQE spectra of the PM-type PPDs cannot match the absorption spectrum of the nanoblend or nanocomposite.

## 4. Conclusions

In this study, oxidative polymerization was used to synthesize a polycrystal-solid form of the poly orthophenylene diamine polymer. Additionally, the sol–gel process was used to create the novel composite made of titanium dioxide nanoparticles [TiO_2_]_NPs_ and poly orthophenylene diamine in powder form. Physical vapor deposition (PVD) technology was used to create thin films of [PoPDA]^TF^ polymer and [PoPDA/TiO_2_]^MNC^ composite measuring 100 ± 3 nm in thickness at room temperature. Using a quartz crystal microbalance, the [PoPDA/TiO_2_]^MNC^ thin film’s thickness was measured using a UNIVEX 250 Leybold machine. The structural properties were investigated using XRD, SEM, and AFM. According to XRD calculations, the typical crystallite sizes of [PoPDA]^TF^ and [PoPDA/TiO_2_]^MNC^ are 102.5 nm and 88.7 nm, respectively. The TD-DFT calculations accurately matched the observed XRD and optical spectra and validated the molecular structure of the examined materials. The values of the optical energy gap 
Egopt
 for [PoPDA]^TF^ and [PoPDA/TiO_2_]^MNC^ are 2.296 eV and 2.114 eV, respectively, in the case of direct transition, and 3.511 eV and 3.209 eV, respectively, in the case of indirect transition. The bandgap energy of the pristine polymer can be reduced by adding [TiO_2_]^NPs^ to the polymer, reducing the bandgap energy by a factor of 51.29%. As established by TD-DFT/DMol^3^, the isolated molecule of the composite [PoPDA/TiO_2_]^Iso^ has a band gap of 2.558 eV. By calculating the values of 
EHOMO
 and 
ELUMO
 energies, which have negative values when the materials are stable, one can theoretically confirm the stability of the manufactured materials. Due to the improvement in the charge transfer brought on by [TiO_2_]^NPs^ nanoparticles, the DC conductivity characteristics of the polymer nanocomposite can be improved by adding these particles. The Au/[PoPDA/TiO_2_]^MNC^/n-Si/Al polymer solar cell can be made using [PoPDA/TiO_2_]^MNC^. The Au/[ PoPDA/TiO_2_]^MNC^/n-Si/Al heterojunction device’s fill factor (FF) and power conversion efficiency are 68.04% and 19.69 at 200 W·cm^−2^, respectively.

## Figures and Tables

**Figure 1 polymers-15-01111-f001:** (**a**) The fabrication process of Au/[PoPDA/TiO_2_]^MNC^/p-Si/Al heterojunction diodes and (**b**) reaction scheme.

**Figure 2 polymers-15-01111-f002:** Combined between the experimental [PoPDA]TF and [PoPDA/TiO_2_]MNC and simulated XRD patterns. Figure (inset) is a 3D Orthorhombic lattice-type and 3D Monoclinic lattice-type for (**a**) [PoPDA]TF and (**b**) [PoPDA/TiO_2_]MNC, respectively.

**Figure 3 polymers-15-01111-f003:** (**a**) Electron density for compounds [PoPDA]^Iso^ and [PoPDA/TiO_2_]^Iso^ as an isolated molecule, (**b**) potential for compounds [PoPDA]^Iso^ and [PoPDA/TiO_2_]^Iso^ as an isolated molecule, and (**c**) MEP of random and block for compounds [PoPDA]^Iso^ and [PoPDA/TiO_2_]^Iso^ as isolated molecule utilizing *TD-DFT/DMOl^3^* programs.

**Figure 4 polymers-15-01111-f004:** SEM images of (**a**) [PoPDA]^TF^ and (**b**) [PoPDA/TiO_2_]^MNC^.

**Figure 5 polymers-15-01111-f005:** AFM images and roughness studies of [PoPDA]^TF^.

**Figure 6 polymers-15-01111-f006:** AFM images and roughness studies of and [PoPDA/TiO_2_]^MNC^.

**Figure 7 polymers-15-01111-f007:** (**a**) UV–Vis absorption measurements of [PoPDA]^TF^ and [PoPDA/TiO_2_]^MNC^ films, (**b**) simulated absorption of [PoPDA]^TF^ and [PoPDA/TiO_2_]^MNC^ using CASTEP.

**Figure 8 polymers-15-01111-f008:** Experimental calculations of bandgap energies for [PoPDA]^TF^ and [PoPDA/TiO_2_]^MNC^ thin films.

**Figure 9 polymers-15-01111-f009:** (**a**) n(λ) and k(λ) spectral dependency per λ, (**b**) simulation via CASTEP optical properties for [PoPDA]^TF^ and [PoPDA/TiO_2_]^MNC^.

**Figure 10 polymers-15-01111-f010:** (**a**) Experimental ε_1_ and ε_2_ versus hν plots, (**b**) ε_1_ and ε_2_ simulation optical properties using CASTEP method in DFT for [PoPDA]^TF^ and [PoPDA/TiO_2_]^MNC^.

**Figure 11 polymers-15-01111-f011:** (**a**) The real (σ_1_) and imaginary conductivity (σ_2_), (**b**) (σ_1_), and (σ_2_) DFT simulation using CASTEP for [PoPDA]^TF^ and [PoPDA/TiO_2_]^MNC^.

**Figure 12 polymers-15-01111-f012:** (**a**–**c**). Application of Wemple–DiDomenico and Sellmeier models on [PoPDA]^TF^ and [PoPDA/TiO_2_]^MNC^.

**Figure 13 polymers-15-01111-f013:** Photoluminescence spectra of [PoPDA]^TF^ and [PoPDA/TiO_2_]^MNC^.

**Figure 14 polymers-15-01111-f014:** Excitation–emission spectra of [PoPDA]^TF^ and [PoPDA/TiO_2_]^MNC^ thin films at 298 K.

**Figure 15 polymers-15-01111-f015:** −log(J)–(V) characteristics curves for Au/[ PoPDA/TiO_2_]^MNC^/n-Si/Al heterojunction diode under dark with T = 298 K and the different illuminations 50 
≤Illuminations ≤
  200 mW/cm^−2^.

**Figure 16 polymers-15-01111-f016:** The dependence of DC resistivity and conductivity of [PoPDA]^TF^ and [PoPDA/TiO_2_]^MNC^ thin films on temperature (1/T).

**Figure 17 polymers-15-01111-f017:** EQE spectra of the PPDs for devices Au/[PoPDA]^TF^/p-Si/Al (A) and nanocomposite Au/[PoPDA/TiO_2_]^MNC^/p-Si/Al (B).

**Table 1 polymers-15-01111-t001:** The experimental and calculated XRD data using the Refine Version 3.0 Software Program (Kurt Barthelme’s and Bob Downs) for [PoPDA]^TF^ and [PoPDA/TiO_2_]^MNC^.

Sample	Observed	Calculated	Δ (Difference)	Debye–Scherrer
2θ	d (Ǻ)	hkl	2θ	d (Ǻ)	2θ	d (Ǻ)	β	D_Av_
[PoPDA]Iso [57]	10.8354	8.09517	101	10.8331	8.09687	−0.0023	−0.0017	1.1058	75.44
ORTHORHOMBIC	15.625	5.63625	021	15.5092	5.67783	−0.1158	−0.04158	0.6446	129.9
a = 13.95(6) Ǻ	18.9173	4.66649	012	18.8681	4.67849	−0.0492	−0.012	1.1225	74.94
b = 13.83(3) Ǻ	21.8087	4.05630	202	21.8518	4.04843	0.0431	0.007869	2.3006	36.73
c = 9.943(9) Ǻ	27.8078	3.19601	141	27.9486	3.18028	0.1408	0.01573	1.4854	57.55
α = β = γ = 90°	34.2042	2.61304	151	34.1377	2.61796	−0.0665	−0.00492	1.327	65.42
V = 1910(8) Ǻ^3^	45.1603	2.00250	414	45.1511	2.00288	−0.0092	−0.00039	0.2562	350.8
rmse = 0.00056	60.7236	1.52200	236	60.727	1.52192	0.0034	0.000077	0.737	130.5
Average			database_code_amcsd 0001206		**102.5**
[PoPDA/TiO_2_]^Iso^ [58]MONOCLINICa = 13.342(1) Ǻb = 13.127(2) Ǻc = 8.994(2) Ǻα = 90^o^, β = 100.28°γ = 90°V = 1550.16(2) Ǻ^3^rmse = 2.08 × 10^−8^	21.9726	4.03012	−311	21.9726	4.03012	0	0	1.54	54.89
30.3608	2.93545	420	30.3608	2.93545	0	0	1.5335	56.07
32.812	2.72197	241	32.812	2.72197	0	0	0.6858	126.1
45.0405	2.00837	−404	45.0405	2.00837	0	0	1.4315	62.75
60.7627	1.52156	821	60.7627	1.52156	0	0	0.6689	143.8
Average	database_code_amcsd 0000913		**88.70**

**Table 2 polymers-15-01111-t002:** Geometry constants for the monomer and dimers [PoPDA]^Iso^ and [PoPDA/TiO_2_]^Iso^ as isolated molecules.

Compounds	E_HOMO_	E_LUMO_	ΔE	χ (eV)	µ (eV)	η (eV)	S (eV)	ω (eV)	ΔNmax	σ
[PoPDA]^Iso^	−4.431	−1.172	3.259	2.802	−2.802	−1.630	−0.307	−2.408	−1.719	−0.614
[PoPDA/TiO_2_]^Iso^	−4.797	−2.239	2.558	3.518	−3.518	−1.279	−0.391	−4.838	−2.751	−0.782

**Table 3 polymers-15-01111-t003:** The optical parameters for the application of Wemple–DiDomenico and Sellmeier models on [PoPDA]^TF^ and [PoPDA/TiO_2_]^MNC^.

Sample	In^direct^(eV)	E_g_^indirect^(eV)	E_o_(eV)	E_d_(eV)	n∞	λo	So	εl	(N/m*)
[PoPDA]^TF^	2.296	3.511	6.10	3.65	1.26	205	14.04	1.93	1.09 × 10^39^
[PoPDA/TiO_2_]^MNC^	2.114	3.209	3.52	1.05	1.14	350	2.44	3.53	1.07 × 10^40^

* is the charge carrier concentration to effective mass ratio.

**Table 4 polymers-15-01111-t004:** Refractive index n(λ), direct and indirect energy gaps 
(ΔE)
  values for nanofiber PoPDA]^TF^ and [PoPDA/TiO_2_]^MNC.^ thin films and comparison with a different polymer.

Film Composition	Symbols	n(λ)	Dir. E_g_	Ind. E_g_	Ref.
Poly(o-phenylenediamine) + poly(p-toluidine)	[PoDA + PpT]^TF^	1.64	2.103	1.88	[49]
Polyethyleneoxide/Carboxymethyl cellulose	PEOCMC (90%)	1.79	2.205	4.03	[81]
	PEO/CMC (80%)	1.97	1.85	3.82
Nanoblend thin films	PoPDA]^TF^	1.26	2.96	3.511	PW
Nanocomposite	[PoPDA/TiO_2_]^MNC^	1.14	2.114	3.209	

Dir. E_g_ (direct energy), Indir. E_g_ (indirect energy), and PW (present work).

**Table 5 polymers-15-01111-t005:** The values of the nonlinear coefficient parameters (*r*_1_ and *r*_2_) for Au/[ PoPDA/TiO_2_]^MNC^/n-Si/Al heterojunction diode.

Temp. (K)	[PoPDA]^TF^	[PoPDA/TiO_2_]^MNC^	Activation Energy Eg0(eV)
	r1	r2	r1	r2	
293	1.97	1.76	2.05	2.93	Eg0 [PoPDA]TF=3.18 eV
313	1.72	1.59	1.78	2.59	Eg0 [PoPDA/TiO2]MNC=2.51 eV
333	1.71	1.51	1.81	2.24	
353	1.29	1.43	1.58	2.62	
373	1.14	1.33	1.49	2.01	

**Table 6 polymers-15-01111-t006:** Au/[ PoPDA/TiO_2_]^MNC^/p-Si/Al polymer device performances under different illumination intensities 
Pin
.

Int. (a)	Vm(b)	Jm(c)	VOC(b)	JSC(d)	Power	FF	η (PCE)
50	0.01976 × 10^−2^	2.95 × 10^−3^	42.025 × 10^−2^	3.01 × 10^−4^	5.83 × 10^−5^	46.05 ± 0.32	11.65 ± 0.51
100	4.787 × 10^−2^	2.91 × 10^−3^	45.238 × 10^−2^	5.40 × 10^−4^	1.39 × 10^−5^	57.02 ± 0.58	13.93 ± 0.48
200	9.558 × 10^−2^	4.12 × 10^−3^	47.952 × 10^−2^	1.21 × 10^−4^	3.94 × 10^−4^	68.04 ± 0.79	19.69 ± 0.67

a = (W·cm^−2^), b = Volt, c = (mA·cm^−2^), and d = (mA·cm^−2^).

## Data Availability

Data are available upon request from the corresponding author.

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
