# Peer review of "RETRACTED: Polymeric Solar Cell with 19.69% Efficiency Based on Poly(o-phenylene diamine)/TiO2 Composites"

_polymers, 2023, doi:10.3390/polym15051111_

Round 1

Reviewer 2 Report

I have following suggestions:

1. Introduction needs some modifications. Why chose POPD and TiO2 nanoparticles for this study? A brief literature review regrading the use of those materials for polymer solar cell application should be included. In addition, many other methods have been reported to form polymer/nanoparticle nanocomposites for solar cell applications, the advantages of the current study should be highlighted. 

Some papers regarding other methods (solution based assembly, in-situ polymerization, etc.) to form polymer/nanoparticle nanocomposites can be cited: Core/shell conjugated polymer/quantum dot composite nanofibers through orthogonal non-covalent interactions. Polymers8(12), p.408.; Effect of CdS nanoparticle content on the in-situ polymerization of 3-hexylthiophene-2, 5-diyl and the application of P3HT‐CdS products in hybrid solar cells. Materials Science in Semiconductor Processing37, pp.259-265.; Core-shell nanocomposite of superparamagnetic Fe3O4 nanoparticles with poly (m-aminobenzenesulfonic acid) for polymer solar cells. Organic Electronics77, p.105462.

2. What's the molecular weight of the synthesized POPD polymer? How to confirm the polymer was successfully synthesized?

3. Figure 1b is not clear, what does the green line between POPD and TiO2 represent?

4. The authors mentioned in the experimental part that FT-IR was measured for the material, but the result is not included in the manuscript.

5. Figure 2 is hard to read. What are the red lines in the spectra?

6. What is the size of the original TiO2 nanoparticles? The authors calculated the crystal size of POPD and POPD/TiO2 to be 10.25 and 8.87, why the crystal size becomes smaller with the addition of the nanoparticles? What kind of interaction is formed between the polymer and nanoparticle?

7. For the SEM images, it's very hard to see TiO2 nanoparticles with the current resolution. I suggest the authors to also conduct TEM measurement of the POPD/TiO2 composite film to study the film morphology and the distribution of nanoparticles within the polymer matrix. 

8. Error bars should be added to the data, such as table 5.

9. Some other questions: Line 17, (xwt%) = (xwtTiO2/wtTiO2+0.5)x100, shouldn't it be (xwt%) = xwtTiO2/(wtTiO2+0.5)x100?; What is MNC in [POPD/TiO2]MNC, what is TF in [POPD]TF?

10. Some English correction is needed. Please make sure there are no grammar and spelling mistakes.  

Reviewer 3 Report

The manuscript entitled “Polymeric solar cell with 19.69% efficiency based on poly(or- 2 tho-phenylene diamine)/TiO2 composites  ” has been submitted by the authors. Some issues to be addressed will improve the quality of the manuscript. Therefore, I recommend this work could be published after the major revision

1.      The author should write down the novelty of this paper.

2.      The English composition requires many improvements. The authors should proofread the manuscript carefully to minimize grammatical errors.

3.      Check the format of the reference and correct all the errors.

4.      In the introduction, the author should like down the comparative study best on this study.

5.      The author needs to increase the inner font size for a clear view to the reader.

Round 2

Reviewer 1 Report

Since authors have done the revision according to the comments, I recommend this paper for the publications. Typos and figure arrangement must do before it is published.

Reviewer 2 Report

The authors have addressed my concerns, and I recommend the acceptance of the manuscript in the present form.

Reviewer 3 Report

The author solve all comments very carefully i recommended to accept in present form.